# 17 Is the New 15: Changing Alcohol Consumption among Swedish Youth

**DOI:** 10.3390/ijerph19031645

**Published:** 2022-01-31

**Authors:** Jonas Raninen, Michael Livingston, Mats Ramstedt, Martina Zetterqvist, Peter Larm, Johan Svensson

**Affiliations:** 1Swedish Council for Information on Alcohol and Other Drugs (CAN), 116 64 Stockholm, Sweden; jonas.raninen@can.se (J.R.); Mats.ramstedt@can.se (M.R.); martina.zetterqvist@can.se (M.Z.); 2Department of Clinical Neuroscience, Karolinska Institutet, 171 77 Stockholm, Sweden; Michael.livingston@curtin.edu.au; 3Unit of Social Work, School of Social Sciences, Södertörn University, 141 89 Huddinge, Sweden; 4Centre for Alcohol Policy Research, La Trobe University, Melbourne, VIC 3086, Australia; 5National Drug Research Institute, Curtin University, Melbourne, VIC 3004, Australia; 6Department of Public Health, Stockholm University, 106 91 Stockholm, Sweden; peter.larm@su.se

**Keywords:** alcohol, youth, survey, Sweden, age of onset

## Abstract

To examine and compare trends in drinking prevalence in nationally representative samples of Swedish 9th and 11th grade students between 2000 and 2018. A further aim is to compare drinking behaviours in the two age groups during years with similar drinking prevalence. Data were drawn from annual surveys of a nationally representative sample of students in year 9 (15–16 years old) and year 11 (17–18 years old). The data covered 19 years for year 9 and 16 years for year 11. Two reference years where the prevalence of drinking was similar were extracted for further comparison, 2018 for year 11 (*n* = 4878) and 2005 for year 9 (*n* = 5423). The reference years were compared with regard to the volume of drinking, heavy episodic drinking, having had an accident and quarrelling while drunk. The prevalence of drinking declined in both age groups during the study period. The rate of decline was somewhat higher among year 9 students. In 2018, the prevalence of drinking was the same for year 11 students as it was for year 9 students in 2005. The volume of drinking was lower among year 11 students in 2018 than year 9 students in 2005. No differences were observed for heavy episodic drinking. The decline in drinking has caused a displacement of consumption so that today’s 17–18-year-olds have a similar drinking behaviour to what 15–16-year-olds had in 2005.

## 1. Introduction

Youth drinking has been declining for two decades in Sweden [1,2]. This development is not isolated to Sweden; similar trends have been observed from several countries such as Australia, Finland and Norway, to name a few [3,4]. Despite an increasing amount of research in recent years, there is currently no explanation for why youth drinking is declining [5,6,7]. Studies from the US, Australia and Sweden have shown a marked increase in the age of onset of alcohol use [8,9,10], indicating that youth today do start to drink but initiate drinking at older ages.

We know from previous studies that the declines in drinking have occurred across all socio-demographic groups [11]. In Sweden, the decline has occurred across all levels of drinkers among 15–16-year-olds [12], 17–18-year-olds [2] and young adults [13]. The declines have also been observed across different drinking patterns [14]. The declines are thus collective within different segments of the population, but less is known about how these changes are moving and interacting across different ages. For example, one study found that changes in youth drinking were disconnected from changes in the adult population’s drinking [1]. Nine out of ten adults in Sweden have consumed alcohol within the last 12 months [15], and roughly three-quarters of the adult population report drinking each month [16]. Alcohol consumption is thus a highly prevalent behaviour in the adult population. Among youth, this is not the case anymore, with non-drinking nowadays being the majority behaviour among 9th graders in Sweden [1]. In comparison with youth from other European countries, Swedish youth report a low drinking frequency but a comparatively high consumption per drinking occasion [17].

The consequences of the declines in youth drinking also remain unclear; a delayed age of onset should be associated with reduced drinking later in life [18]. One study, using age-period-cohort modelling, found that whereas drinking declined among those 16-year old or younger, there was no decrease in drinking among 18-year-olds [19]. This indicated that the observed declines in youth drinking were isolated to certain ages and that young people ‘catch up’ with previous generations in their drinking behaviour as they grow older [19]. Similarly, a longitudinal study from Australia found that the consumption level among more recent cohorts was significantly lower compared to previous cohorts of adolescents and young adults, but the rate of increase in drinking became significantly faster as the more recent cohorts aged [20].

Several studies have shown contemporary and concordant changes in parenting practices and have shown that these changes are associated with the decline in drinking [3,21,22]. There are also reports of changes in the parent–child relationship, with youth nowadays spending more time with parents and conforming to their parents’ lifestyle [23]. One of the key theories put forward by researchers to explain this decline is that it represents a broader shift of ‘delayed adolescence’ or ‘prolonged adolescence’ [6,24,25]. Few alcohol researchers have, however, explicitly examined this question across a broad range of alcohol use indicators, i.e., beyond simple measures of abstinence or drinking volume.

This study aims to examine and compare trends in alcohol consumption among Swedish 9th graders and Swedish 11th graders using nationally representative school surveys to see if trends show similar developments for these two age groups. A further aim is to compare the drinking patterns and self-reported harm in the two age groups to examine if there has been an upward shift in drinking habits as a result of the declines in youth drinking.

## 2. Materials and Methods

Data on youth alcohol consumption was retrieved from a database collected by The Swedish Council for Information on Alcohol and other Drugs (CAN). CAN have conducted school surveys of alcohol consumption among year 9 students (15–16 years of age) annually since 1971. Since 2004, they have also conducted surveys with year 11 students (aged 17–18 years). The samples are nationally representative of students in the two grades. The sampling, conducted by Statistics Sweden, uses a stratified approach to ensure that all regions in Sweden are represented. School class rather than pupil is used as the unit when drawing the sample, i.e., if a class is drawn, then all students in that class fill in the questionnaire. The response rates in year 9 vary between 90 and 83% and, for year 11, between 86 and 80% [26]. The survey is an anonymous, paper and pen questionnaire that is completed in the classroom [26].

The alcohol questions used in this study consisted of:(1)‘Have you ever had a drink containing alcohol?’—those reporting zero drinking occasions were categorised as non-drinkers.(2)A quantity and frequency scale for the last twelve months. These questions were asked separately for each beverage type (beer/wine/spirits/cider/alcopops) and then summarised into a measure of overall drinking in litres of 100% alcohol during the last twelve months.(3)Heavy episodic drinking (HED) was measured by the question: ‘How often do you drink alcohol corresponding to at least half a bottle of spirits or one bottle of wine or four large bottles of strong cider or four cans of strong beer during one drinking session?’. The response categories were ‘some times per week’, ‘some times per month’, ‘about once a month’, ‘a few times per year’, ‘more seldom’ and ‘never’. This was dichotomised into monthly or more often = 1 and less than monthly = 0.(4)Alcohol-related harm was measured using the question: Have you ever experienced any of the following problems due to your drinking of alcohol? Quarrel/Accident or injury. The response options were ‘Never’, ‘Once’, ‘twice’ or ‘three times or more’ and were dichotomised into 0 = Never and 1 = once, or more than once.

### Analytical Approach

In the most recent survey available to us (2018), the prevalence of drinking among 11th graders was 72.5 per cent, we then scanned the trend for the 9th graders to find a suitable year for comparison. In 2005, the prevalence of drinking among 9th graders was 71.8 per cent, and this year was then chosen as our reference year for further comparison. Data from 2005 was used for year 9 (*n* = 5423) and data from 2018 for year 11 (*n* = 4878). Detailed data on drinking were extracted for the two years. These were then used to compare the two years with respect to all the measures of drinking. In the final sample, 49.8% were female. Descriptive comparisons were made between the two years. Differences were tested using chi-square and *t*-tests. The significance level was set at *p* < 0.05 for all analyses. Cluster robust standard errors were used to control for the respondents being nested within school classes.

## 3. Results

Figure 1 displays the trends in the prevalence of drinking for both age groups. Among 9th graders, the prevalence of drinking dropped from 81 per cent in 2000 to 39 per cent in 2018. The average rate of change is two percentage points per year across the entire period, and this rate of change has been relatively stable. Among 11th graders, the prevalence of drinking dropped from 89 per cent in 2004 to 72 per cent in 2018. The rate of change has been one percentage point per year across the entire period. Drinking has declined for both age groups but with a somewhat more accelerated trend among 9th graders.

In Table 1, we compare drinking characteristics of 11th graders in 2018 and 9th graders in 2005. Small, non-significant differences were observed between the grades in heavy episodic drinking. The prevalence of high consumers and the average volume of pure alcohol displayed larger differences that were statistically significant. The volume of consumption was 3.5 L for 9th graders in 2005 and 2.8 L for 11th graders in 2018. In both age groups, 11 per cent reported having had an accident when they had been drinking and 22 per cent of 9th graders reported having been in a quarrel in 2005, whilst 19 per cent of 11th graders reported that in 2018. Neither of these differences were significant.

## 4. Discussion

It is important to understand the consequences of the marked declines in youth drinking seen over the past two decades. Our results show that drinking has declined both among 9th graders and 11 graders but with a steeper decline among the 9th graders. As a consequence of the declining trends, the prevalence of drinking in 2018 among 11th graders was the same as the prevalence of drinking among 9th graders in 2005. We found striking similarities in drinking behaviour between Swedish 9th graders in 2005 and Swedish 11th graders in 2018, with the only significant difference found for the average volume of consumption. This measure was found to be significantly lower among the older year 11 students in 2018 compared to the 9th graders in 2005. In general, our results indicate that there has been a shift in the drinking habits of Swedish youth so that the drinking habits have been ‘pushed’ for two years. In a wider sense, the results imply that the overall prevalence of drinking in a generation seems to be an indicator of the drinking habits and drinking patterns, rather than there being a certain drinking pattern decided by biological age. Our results also show that the self-reports of experiencing alcohol-related harms are similar in the two age groups during years with similar drinking patterns. Previous studies have shown that the association between drinking and harm has been stable among Swedish 9th graders during the period 1995–2012, even though consumption has fluctuated during this period [27].

These results mirror those of other studies examining the age of onset [8,9], which have shown an increasing mean age of onset of drinking in recent years. Our results are however somewhat in contrast to those of Lintonen et al. [19], who showed that there was no continuation of the decline in drinking among Finnish youth, i.e., the declines were only observed among younger youth, whilst at the age of 18, older and newer cohorts drank the same. Our results show that the declines in drinking observed among Swedish youth have caused a displacement of drinking, so 17–18-year-olds in 2018 drink in a similar manner as 15–16 year-olds did twelve years prior. This would fit with the notion of a ‘prolonged adolescence’, where youth today start with adult behaviours at an older age [6,24].

A lower age of onset has been linked to both higher levels of drinking later in life and a higher probability of alcohol-related harms [18,28,29]. Increasing the age of onset of drinking has therefore been a public health priority to reduce alcohol-related harms. Our results give support to the basic idea of pushing the age of onset upwards as beneficial from a public health perspective. It is important to further our understanding of the factors behind the declines in youth drinking so that we can implement policies that support this development.

Our results suggest that the drinking habits have been pushed to later ages, rather than young people persistently abstaining from alcohol. Children and young people usually do not drink alcohol, and starting to drink is part of entering adulthood [30]. Thus, young people that do not drink have most likely not chosen to abstain from alcohol but rather have simply not initiated the behaviour. Our results indicate that the trends observed in youth drinking in Sweden are more a matter of a displacement of behaviour to later ages than an altogether increase in ongoing abstinence. In trying to understand this development, we should also be looking for factors and explanations for why young people initiate drinking at older ages today, in addition to why they do not drink.

There are some limitations that need to be kept in mind when interpreting these results. The results are all based on self-reported survey information, which is usually underestimated when it comes to alcohol consumption [31]. Self-reported drinking has, however, been shown to be reasonably reliable among adolescents [32]. The issue of underestimation should also be less salient in the current study as the questions and mode of data collection have not changed during the study period, so possible underestimation should be the same across all years. The samples are nationally representative of students in year 9 and year 11 since mandatory school ends after year 9 in Sweden and not all continue to upper secondary school. It is plausible that those not continuing after year 9 would belong to a group with an increased risk of drinking. Again, the possible effects of this should be the same for all years, and drinking has declined also among high-risk groups of Swedish youth [33], but the results presented here are limited to year 11 students and not all 17–18 year-olds.

Our results show that there has been a shift in the drinking habits of Swedish youth, but our analyses are limited to different cross-sectional samples of youth, meaning that we know little about individual drinking trajectories. There is a paucity of longitudinal studies in this area [18] and a need for longitudinal examinations of the importance of the timing of onset of drinking for later drinking habits.

Today’s youth seem to belong to a generally well-behaved generation as there are observations of declines in crime [34] and other risk behaviours [35] parallel to the declines in youth drinking. There are also reports of normalising non-drinking in this generation [36,37], with small or no differences seen between drinkers and non-drinkers [38]. There is also some evidence that today’s youth are more health-conscious [39,40]. Since risk factors and risky behaviours tend to cluster together and co-occur [41], it will be interesting to examine what this means for the drinking habits of today’s youth in the long term. Generational differences, or cohort effects, have previously been observed when it comes to drinking habits [42,43], and future research should examine if the upwards shift towards drinking at older ages reported here will translate into a cohort effect impacting the drinking habits of this generation across their life-course.

The major strength of the present study is the high-quality data, with data being collected in a consistent manner by the same organisation and the same mode of collection across the entire study period. The response rates are high and in line with the average for all countries included in the latest wave of the European School Survey Project on Alcohol and other Drugs (ESPAD) [17]. Furthermore, the sample is large and nationally representative. This means that the findings are likely robust and not artefacts due to reliability or validity problems or diverging trends in specific sub-groups of the population.

## 5. Conclusions

In 2018, Swedish year 11 students had the same drinking behaviours as year 9 students in 2005. The declines in youth drinking have shifted drinking to later in life so that today’s 17–18-year-olds drink about the same as 15–16-year-olds did twelve years ago. The long-term importance of this needs to be studied further.

## Figures and Tables

**Figure 1 ijerph-19-01645-f001:**
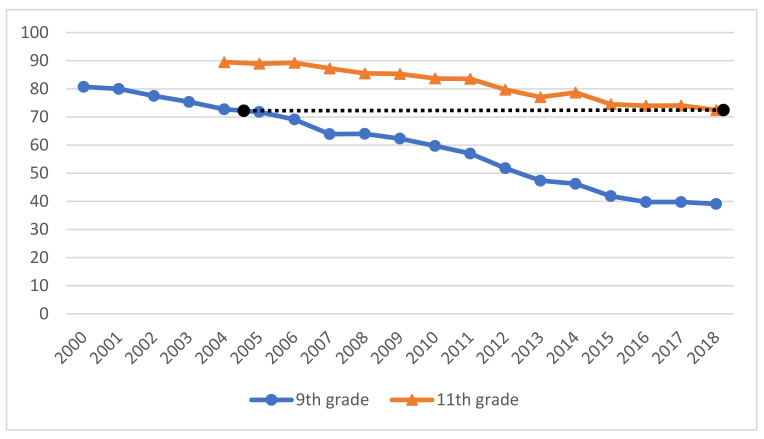
Prevalence of alcohol consumers in year 9 and year 11.

**Table 1 ijerph-19-01645-t001:** Comparison of measures of drinking: 9th grade students in 2005 and 11th grade students in 2018.

	9th Grade 2005 (*n* = 5423)	11th Grade 2018 (*n* = 4878)	Pearson Chi-Square	DF	*p*-Value
Drinkers (%)	71.8	72.5	2.7894		0.3126
Average volume (m)	**3.5**	**2.8**	**22.21 ***		**<0.001**
Heavy episodic drinking (%)	24.1	23.7	0.2709		0.7263
Had an accident (%)	11.2	10.9	0.3705		0.6172
Quarrel (%)	22.0	19.2	7.2611		0.0663

Bold text indicates significant difference (*p* < 0.05) between 9th graders and 11th graders. * *t*-test statistics.

## Data Availability

The data presented in this study are available on request from the corresponding author. The data are not publicly available due to privacy and ethical restrictions.

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
