# Peer review of "17 Is the New 15: Changing Alcohol Consumption among Swedish Youth"

_ijerph, 2022, doi:10.3390/ijerph19031645_

Round 1

Reviewer 1 Report

This study intended to examine and compare trends in drinking prevalence in nationally representative samples of Swedish 9th and 11th grade students between the years 2000 and 2018. A further aim is to compare drinking behaviors in the two age groups during years with similar drinking prevalence. The manuscript has several strengths, but this reviewer has a few major comments and a number of minor comments. Primarily, authors need to strengthen the methods and results sections.

Line 15 – In the abstract, you have indicated that the “Data 15 were drawn from annual surveys of a nationally representative sample of students”. Please provide the name of the survey.

Line 25 – What is HED? I assume this is heavy episodic drinking, but please expand the abbreviation.

Line 21 and 25 – You stated “The prevalence of drinking declined in both age groups during the study period.” And then wrote “Drinking has declined in both age groups during the entire study period.” Is this redundant or are you writing about two different findings?

Line 42 – Please correct numerous grammatical and spelling errors, for example, “dis-connected” should be without hyphen, and “adult populations drinking” needs to be corrected.

How was the parental consent and child assent obtained in this study? In the end of the article, you have indicated “Informed Consent Statement: Not applicable”. What does it mean? Wasn’t the ethical approval obtained in this study? Or, did you mean that it was not obtained for the secondary data analysis?

Line 90 – What is the basis for dichotomizing into monthly or more often and less than monthly? I understand that you did this for analysis, but I’m asking about the cutoff. Is this based on a theoretical concept, empirical finding, or just a post-hoc decision? Similarly, justify the basis of dichotomizing alcohol related harm.

Table 1 – Include the sample size in the title or in each column heading.

Line 103 – “In the final sample 49.8 % were female.” Are you referring to the total sample (2005 and 2018) or all years together (2000 thru 2018)?

Authors have conducted a trend analysis for 2000 – 2018. However, no description has been provided for data from those years. Statistical analysis does not include a description of the trend analysis. Did you do any adjustments? Did you use sampling weights to make the years comparable?

Were there any missing data? How did you handle missing data?

Figure 1 – What is 2012A and 2012B. If this is a special sample, please describe it in a footnote.

Please provide sample characteristics for each year (2005 and 2018). Include the demographic characteristics of the sample for each year.

Line 119 – 126 – Authors have presented several findings, both significant and non-significant. You need to provide the test statistic (e.g., chi-square), degrees of freedom (if applicable), and the p-value for each of these. Also, include a column to Table-1 to indicate the statistical significance.

Discussion is well-written. However, authors need to add more citations, for example, for numerous claims in the third paragraph of the discussion.

Author Response

Author's Reply to Reviewer

Comments and Suggestions for Authors

This study intended to examine and compare trends in drinking prevalence in nationally representative samples of Swedish 9th and 11th grade students between the years 2000 and 2018. A further aim is to compare drinking behaviors in the two age groups during years with similar drinking prevalence. The manuscript has several strengths, but this reviewer has a few major comments and a number of minor comments. Primarily, authors need to strengthen the methods and results sections.

Line 15 – In the abstract, you have indicated that the “Data 15 were drawn from annual surveys of a nationally representative sample of students”. Please provide the name of the survey.

Response: This description is given in the Materials and Methods section on lines 69-72 where also a reference is given for the detailed reports for the survey (reference #23). We feel that this detailed information is redundant in the abstract and would only take up the scarce and limited space provided in the abstract.

Line 25 – What is HED? I assume this is heavy episodic drinking, but please expand the abbreviation.

Response: Thank you for pointing this out. We have now expanded the abbreviation in the abstract and also noted the abbreviation the first time it is used in the text on line 87.

Line 21 and 25 – You stated “The prevalence of drinking declined in both age groups during the study period.” And then wrote “Drinking has declined in both age groups during the entire study period.” Is this redundant or are you writing about two different findings?

Response: Thank you for noticing this. We have now changed the text so that the second sentence reads “The decline in drinking has caused a displacement of consumption so that today’s 17-18-year-olds have a similar drinking behaviour like 15-16-year-olds had in 2005.”

Line 42 – Please correct numerous grammatical and spelling errors, for example, “dis-connected” should be without hyphen, and “adult populations drinking” needs to be corrected.

Response: We have now changed this and language edited the entire text.

How was the parental consent and child assent obtained in this study? In the end of the article, you have indicated “Informed Consent Statement: Not applicable”. What does it mean? Wasn’t the ethical approval obtained in this study? Or, did you mean that it was not obtained for the secondary data analysis?

Response: In Sweden parental consent is not needed for respondents 15 years or older. The respondents themselves provide a passive consent when choosing to participate in the study. They are provided information that participation is voluntary and that by filling out the questionnaire they consent to take part in the survey. No active consent form is however filled out. The survey is done in accordance with the Helsinki declaration, Swedish law and ethical guidelines for research. Ethical approval was not obtained for the secondary data analysis done within the framework of the present paper. 

Line 90 – What is the basis for dichotomizing into monthly or more often and less than monthly? I understand that you did this for analysis, but I’m asking about the cutoff. Is this based on a theoretical concept, empirical finding, or just a post-hoc decision? Similarly, justify the basis of dichotomizing alcohol related harm.  

Response: The dichotomizing of HED into monthly or more often is a standard way of analysing and reporting this. This is done in the same way in the national reports based on the same data (see for example reference #23) and also in the European School survey Project on Alcohol and other Drugs (ESPAD). This is also used in other publications on youth drinking (see for example reference #3). The basis for dichotomizing alcohol related harm into any versus none was a post-hoc decision.  

Table 1 – Include the sample size in the title or in each column heading.

Response: Done.

Line 103 – “In the final sample 49.8 % were female.” Are you referring to the total sample (2005 and 2018) or all years together (2000 thru 2018)?

Response: This is for the 2005 and 2018 data.

Authors have conducted a trend analysis for 2000 – 2018. However, no description has been provided for data from those years. Statistical analysis does not include a description of the trend analysis. Did you do any adjustments? Did you use sampling weights to make the years comparable?

Response: We have not conducted a trend analysis, we simply report the trends. A description of the data, valid for all years, are provided in the first paragraph of the Materials and Methods section. The survey uses post-stratification weights each year to adjust for possible skewness in the data regarding sex and geographical region in Sweden.

Were there any missing data? How did you handle missing data?

Response: For the alcohol questions (lifetime, volume and HED) the missing data is very low, 1.12 % for HED is the most. For the harm questions the missing is somewhat higher, around 7%. Respondents with missing data were excluded from the analysis and no adjustment or imputing was done.   

Figure 1 – What is 2012A and 2012B. If this is a special sample, please describe it in a footnote.

Response: Thank you for pointing this out. In 2012 a split-half was done when changes were made to some of the questions. Only one of these should be included when reporting the trends. We have now updated the figure.

Please provide sample characteristics for each year (2005 and 2018). Include the demographic characteristics of the sample for each year.

Response: This is done in the final paragraph of the materials and methods section on lines 104-105. We do not have access to any further demographic information, other than the respondents sex.

Line 119 – 126 – Authors have presented several findings, both significant and non-significant. You need to provide the test statistic (e.g., chi-square), degrees of freedom (if applicable), and the p-value for each of these. Also, include a column to Table-1 to indicate the statistical significance.

Response: This is now included.

Discussion is well-written. However, authors need to add more citations, for example, for numerous claims in the third paragraph of the discussion.

Response: Thank you. We have now re-written this paragraph and softened some of our claims somewhat and we have also provided references where possible.

Reviewer 2 Report

This is an important social topic, and interesting as well. Please, do note the comments below:

In line 32, give the names of some of these several countries you talk about.

The introduction does not give sufficient background on alcohol consumption in the case study area i.e Sweden

In the methodology, since the data is not directly collected by the authors but are based on a secondary national data, it is expected to give reliable citation to this source of the data as the accuracy, or otherwise is not in the domain of the authors.

Since this is only based on 2005, and 2018, and given that we are in 2022, how can we be conclusive of the assertion that “ the declines in drinking observed among Swedish youth has caused a displacement of drinking so that today’s 17-18-year-olds drink in a similar manner as 15-16 year-olds did twelve years prior.” This is because, the data is four years already in 2022, and therefore this might not be reflexive of current alcohol consumption habits especially given the outbreak of Corona from 2020 with its lockdown, and other changes to previously normal-social life which equally has an influence on alcohol consumption habits/ patterns. Given that 2018, is the recent year available, what measures can be made from the previous years presented to guide a projection that can scientifically support a reasoning that the identified trend is more likely to be reflexive or similar four years on? Consider this

Again, if we are to go by this assertion, what implications does it have, and what policy implications does the study give? This is a social issue and there has to be policy implications

Also, one will expect to see what factors had accounted in this result in 2005, and 2018 among the 9th graders, and 11 graders respectively. This the methodology needs to delve into. 

Author Response

Author's Reply to Reviewer

Comments and Suggestions for Authors

This is an important social topic, and interesting as well. Please, do note the comments below:

In line 32, give the names of some of these several countries you talk about.

Response: Done. The text now reads “This development is not isolated to Sweden, similar trends have been observed from several countries like Australia, Finland and Norway to name a few”

The introduction does not give sufficient background on alcohol consumption in the case study area i.e Sweden

Response: We do think that the second paragraph gives a fairly detailed description of alcohol consumption and how this has changed. We are happy to add further information but could you please be more specific in what information you are missing?

In the methodology, since the data is not directly collected by the authors but are based on a secondary national data, it is expected to give reliable citation to this source of the data as the accuracy, or otherwise is not in the domain of the authors.

Response: This is provided in the first paragraph of the Materials and Methods section on line 78, reference #23.

Since this is only based on 2005, and 2018, and given that we are in 2022, how can we be conclusive of the assertion that “ the declines in drinking observed among Swedish youth has caused a displacement of drinking so that today’s 17-18-year-olds drink in a similar manner as 15-16 year-olds did twelve years prior.” This is because, the data is four years already in 2022, and therefore this might not be reflexive of current alcohol consumption habits especially given the outbreak of Corona from 2020 with its lockdown, and other changes to previously normal-social life which equally has an influence on alcohol consumption habits/ patterns. Given that 2018, is the recent year available, what measures can be made from the previous years presented to guide a projection that can scientifically support a reasoning that the identified trend is more likely to be reflexive or similar four years on? Consider this

Response: Thank you for this important comment. We do think it is plausible that the Corona pandemic has had an effect on youth drinking. However, we do not want to speculate on this as we feel that this is somewhat outside the scope of the present paper. We have instead changed the text so that it now reads “Our result rather show that the declines in drinking observed among Swedish youth has caused a displacement of drinking so that 17-18-year-olds in 2018 drink in a similar manner as 15-16 year-olds did twelve years prior.” Rather than stating that “today’s 17-18 year olds” do so.

Again, if we are to go by this assertion, what implications does it have, and what policy implications does the study give? This is a social issue and there has to be policy implications

Response: We have now added a paragraph in the discussion on this. “A lower age of onset has been linked to both higher levels of drinking later in life and a higher probability of alcohol related harms (15; 25; 26). Increasing the age of onset of drinking has therefore been a public health priority to reduce alcohol related harms. Our results give support to the basic idea of pushing the age of onset upwards as beneficial from a public health perspective. It is important to further our understandings of the factors behind the declines in youth drinking so that we can implement policies that support the development.”  

Also, one will expect to see what factors had accounted in this result in 2005, and 2018 among the 9th graders, and 11 graders respectively. This the methodology needs to delve into. 

Response: Apologies, but we do not understand this comment or what we are expected to do.

Reviewer 3 Report

Raninen and colleagues present trends in drinking in nationally representative samples of Swedish adolescents in the 9th and 11th grade. Examining data from 2000 to 2018, the authors find that adolescents are not drinking less, per se, but instead appear to be delaying the onset of alcohol consumption. That is, drinking prevalence rates amongst adolescents in the 11th grade resemble those of their younger counterparts (9th grade) in previous waves of this survey.

Overall, the paper is excellently and engagingly written. The methodology and analyses are appropriate, and the interpretation of results is expertly done. I only have a few minor points, mostly typographical errors:

  • In the abstract on line 14, 11 should be 11th.
  • In the introduction, line 35, has should be have.
  • Line 169, "the results" should be "these results".
  • Line 200, the authors are correct that the high response rate for this survey is a strength of their work. They may wish to cite response rates in similar surveys in other countries to provide context.

Author Response

 Author's Reply to Reviewer

Comments and Suggestions for Authors

Raninen and colleagues present trends in drinking in nationally representative samples of Swedish adolescents in the 9th and 11th grade. Examining data from 2000 to 2018, the authors find that adolescents are not drinking less, per se, but instead appear to be delaying the onset of alcohol consumption. That is, drinking prevalence rates amongst adolescents in the 11th grade resemble those of their younger counterparts (9th grade) in previous waves of this survey.

Overall, the paper is excellently and engagingly written. The methodology and analyses are appropriate, and the interpretation of results is expertly done. I only have a few minor points, mostly typographical errors:

  • In the abstract on line 14, 11 should be 11th.
  • In the introduction, line 35, has should be have.
  • Line 169, "the results" should be "these results".
  • Line 200, the authors are correct that the high response rate for this survey is a strength of their work. They may wish to cite response rates in similar surveys in other countries to provide context.

 Response: Thank you. These have all been corrected.

Round 2

Reviewer 1 Report

Thanks for addressing my comments. I have no further comments.

Author Response

Thank you for the opportunity to further revise our manuscript and the detailed and thoughtful suggestions. We have now revised the text in accordance with the suggestions. We have changed the first sentence in the second paragraph and also added a new section to the second paragraph detailing the drinking habits in Sweden further. This now reads. “Nine out of ten adults in Sweden have consumed alcohol during the last 12 months [15] and roughly three quarters of the adult population report drinking each month [16]. Alcohol consumption is thus a highly prevalent behaviour in the adult population. Among youth this is not the case anymore, with non-drinking nowadays being the majority behaviour among 9th graders in Sweden [1]. In Sweden the legal age to purchase alcohol is 18 for on-premise outlets and 20 for off-premise outlets. In comparison with youth from other European countries, Swedish youth report a low drinking frequency but a comparatively high consumption per drinking occasion [17].”

Reviewer 2 Report

Thank you for your response to the initial comments: 

However, I think the comments had not been adequately addressed and can therefore not recommend the paper for publication at this point. This is particularly so due to the fact that the data for the work is too old to pass a novelty test in the present time. Neither did the work sufficiently made rigorous analysis from the presented years that could form a basis to make a simulation into the current year for the conclusion drawn.

Clarification on Some of the Comments:

In comment 2, what I meant was that the work consider and delve more into the drinking habits in Sweden itself in addition to other country scenarios that paragraph 2 presented.

Making a rigorous analysis of past incidence to form basis for a simulation into the future is not speculation in research work. That is why your analysis needs to be rigorous enough to form a good justification why you say a situation is likely to remain same, or change in the future date. 

Finally, what I meant in the last comment is that readers will expect to see what have been the contributory factors to the results found for the 2005, and 2018. This is however missing. Given the insufficiency, and old nature of the data used in this study, my recommendation will be to delve some of these issues to compliment the richness of the data, and to help make a rigorous scientific argument in the paper.  

Author Response

(The authors gave the same response as above.)
